physiology/biomechanics

remora, adhesive disc, mechanoreceptor, attachment, touch

**Author for correspondence:**
Brooke E. Flammang
e-mail: flammang@njit.edu

# Knowing when to stick: touch receptors found in the remora adhesive disc

Karly E. Cohen[1,3], Brooke E. Flammang[2], Callie H. Crawford[2] and L. Patricia Hernandez[1]

[1]Biology Department, University of Washington, Life Sciences Building, Seattle, WA 98195, USA
[2]Department of Biological Sciences, New Jersey Institute of Technology, University Heights, Newark, NJ 07102, USA
[3]Department of Biological Sciences, The George Washington University, Science and Engineering Hall, Suite 6000, Washington, DC 20052, USA

(iD) BEF, 0000-0003-0049-965X

Remoras are fishes that piggyback onto larger marine fauna via an adhesive disc to increase locomotor efficiency, likelihood of finding mates and access to prey. Attaching rapidly to a large, fast-moving host is no easy task, and while research to date has focused on how the disc supports adhesion, no attention has been paid to how or if remoras are able to sense attachment. We identified push-rod-like mechanoreceptor complexes embedded in the soft lip of the remora adhesive disc that are known in other organisms to respond to touch and shear forces. This is, to our knowledge, the first time such mechanoreceptor complexes are described in fishes as they were only known previously in monotremes. The presence of push-rod-like mechanoreceptor complexes suggests not only that fishes may be able to sense their environment in ways not heretofore described but that specialized tactile mechanoreceptor complexes may be a more basal vertebrate feature than previously thought.

## 1. Background

Remoras (family Echeneidae) are well known for their hitch-hiking behaviour; these fishes attach to other organisms such as sharks, whales, billfish and turtles [1–3] to take advantage of greater locomotor efficiency, increased probability of meeting mates, reduced predation and access to food such as parasites living on the host. The remora adhesive disc is capable of attaching to surfaces of varying roughness and stiffness and remaining attached under high shear conditions, where drag forces impart pressure strain-inducing sliding relative to the host [4,5]. Derived from dorsal fin elements [6,7], the adhesive disc is comprised of a central series of flat pectinated, or comb-like, lamellae surrounded by a

**Figure 1.** Putative push-rod mechanoreceptor complexes within the remora disc lip. (*a*) Dorsal view of remora adhesive disc attached to a glass surface. *fl*, fleshy lip of disc; *la*, lamella; *sp*, spinules. (*b*) Putative mechanoreceptor complexes in the anterior lip denoted by white arrows. (*c*) Putative mechanoreceptor complexes in the posterior lip denoted by white arrows. (*d*) Schematic overview of putative push-rod mechanoreceptor complex distribution. (*e*) Inset highlights a lateral view of an individual putative push-rod cell complex.

fleshy lip (figure 1). Adhesion is accomplished by the remora coming into contact with the host surface and rotating the lamellae; this both engages the small spinules of the lamellae to generate friction and creates a subambient pressure under the disc resulting in a suction seal of the fleshy lip [8,9]. Because remoras attach to rapidly swimming hosts, the adhesive mechanism must engage immediately upon contact with the host organism to achieve attachment.

Over the past few decades, a small number of studies have investigated proprioception in fishes [10–12]. Mechanoreceptor complexes that respond to touch were once thought to be absent in teleosts [13,14], as fishes sense their environment predominantly through a lateral line canal system on their head and body [10] that is sensitive to water flow and vibrations in the environment [15]. Specific pressure feedback would seem imperative for remoras to know when they have achieved contact and then initiate attachment. Presumably, pressure feedback would be important to any fishes that physically interact with a solid environment. High densities of free nerve endings within the epidermis of fishes [11,12,16,17] are hypothesized to respond to tactile cues. However, more specific tactile reception exists as well. Strain receptors in fish fins are able to sense hydrodynamic loading and contact with surrounding surfaces within a cluttered environment [18–20]. Additionally, moray eels use 'touch-corpuscles' in addition to free nerve endings to search out appropriate prey items [21]. Importantly, however, the tactile receptors in moray eels are not homologous to touch receptors found in other vertebrates, as indicated by phylogenetic distribution and differing in both anatomy and function.

Specialized touch receptor complexes have not yet been identified in fishes, but they are common in amphibians and amniotes [22], including those that are secondarily aquatic [23,24]. Toothed whale calves, for example, in addition to the somatosensory proprioception of their skin, have short vibrissae, or small hairs, around their mouth to let them know they are in contact with their mother to nurse [25]. One additional type of touch receptor complex has been identified in mammals that habitually search for food in murky water or root in soil: monotremes, such as the platypus and echidnas, have a unique type of push-rod mechanoreceptor complex on their bill [26–28]. The push-rod mechanoreceptor complex is a column of tightly associated cells that projects upwards elevating the skin surface and is associated with

myelinated stem axons and vesicle chain receptors at its base [26,29]. Mechanical stimulation or shear stress at the epidermis induces movement of the push-rod, which generates anisotropic sensory information [30].

Herein, we describe a putative push-rod mechanoreceptor complex found for the first time, to our knowledge, in fishes, in the epithelium of the remora adhesive disc, and compare its morphology to that found across monotremes, the other organisms primarily associated with this type of mechanoreceptor. Using various histological methods, we identified several sensory structures including putative push-rod mechanoreceptor complexes along the distal edges of the soft tissue lip surrounding the adhesive disc. The presence of push-rod mechanoreceptor complexes would allow remoras to immediately sense when they have made contact with a host and rapidly initiate attachment as well as sense shear forces and modulate attachment for long-term adhesion.

# 2. Methods

## 2.1. Sample collection

*Echeneis naucrates*, sharksucker remora, were maintained at the New Jersey Institute of Technology (NJIT) Department of Biological Sciences aquatic animal facility and euthanized and fresh-frozen following NJIT/Rutgers University IACUC protocol 17058-A0-R1. This species was chosen because it is the most generalized and cosmopolitan in its host association as compared to other remora species [1,2]. Therefore, it adheres to the widest range of substrates, thus being most representative of other remora species as well as ruling out very specialized morphology-substrate interactions. In addition, this species was already housed at NJIT and is the only one possible to get live from commercial suppliers whereas other species are typically not easily collected or carried for live sale. Freezing of specimens was necessary to transfer them from NJIT to the University of Washington and George Washington University, where histological and scanning electron microscopy (SEM) studies were conducted.

## 2.2. Morphological observation

The discs of each known species of remora were observed under a dissecting microscope to quantify abundance of putative push-rod mechanoreceptor complexes along the disc lip. Specimens included *Phtheirichthys lineatus* (MCZ 33448, MCZ 83211), *Echeneis naucratoides* (MCZ 8678, MCZ 172399), *E. naucrates* (MCZ 8663 and one specimen prior to histological sectioning), *Remora albescens* (MCZ 31364), *Remora australis* (MCZ 8685), *Remora remora* (MCZ 83212), *Remora osteochir* (MCZ 43246) and *Remora brachyptera* (MCZ 8668).

## 2.3. Computed microtomographic scanning

The head and disc of one remora was fixed in 10% formalin, stored in 70% ethanol (EtOH), and computed microtomographic (μCT) scanned at 125 uA, 80 kV, with 47 ms exposure time, with a voxel size of 31 μm in a Bruker Skyscan 1275 (Microphotonics, Inc., Allentown, PA, USA) located in the Otto York Bioimaging facility at the NJIT, as a comparison to rule out calcified tissue with respect to stained soft tissue. The specimen was then transferred to 3% phosphotungstic acid (PTA) for three weeks and scanned again at 125 uA, 80 kV, with 46 ms exposure time with a 1 mm aluminium filter, with a resulting voxel size of 15 μm. Scanned images were reconstructed with NRecon reconstruction software (Bruker) and measurements made using MIMICS RESEARCH SUITE v. 20.0 (Materialise, Leuven, Belgium). A *t*-test was used to compare the abundance of putative push-rod mechanoreceptor complexes in the anterior and posterior regions of the disc of *E. naucrates* ($n = 3$).

## 2.4. Scanning electron microscopy

Evenly spaced sections of the anterior, middle and posterior lip were dissected ($n = 9$ samples from three individuals) for SEM. Additional samples were taken of the lamellae of the adhesive disc to see if there were any mechanoreceptors in the epithelium of the lamellae ($n = 6$ samples). The lip was prepped in one of two ways. Previously frozen tissue was fixed in 10% buffered formalin and transferred to glutaraldehyde before dehydrating to 100% EtOH. The second protocol called for thawing the remora lip and placing it overnight in $dH_2O$. This was done to keep structures from deforming under the desiccation of the rest of the SEM preparation. Similar to samples that were fixed in formalin, samples were post-fixed in glutaraldehyde and brought through an ethanol dehydration series to 100% EtOH.

**Table 1.** Distribution of putative push-rod mechanoreceptor complexes in remora discs. (Habitat and host association information from [1,2].)

| species | push-rod distribution (cm$^{-2}$) | | habitat | host associations |
| | anterior | posterior | | |
| --- | --- | --- | --- | --- |
| *Phtheirichthys lineatus* | 15–24 | 11–13 | reef | reef fishes, sharks |
| *Echeneis naucrates* | 9–12 | 2–4 | reef | reef fishes, sharks |
| *Echeneis naucratoides* | 8–14 | 2–6 | reef | sharks |
| *Remora albescens* | 6–12 | 4–8 | pelagic | rays |
| *Remora australis* | 7–13 | 5–7 | pelagic | cetaceans |
| *Remora remora* | 17–24 | 11–15 | pelagic | sharks |
| *Remora osteochir* | 14–20 | 8–18 | pelagic | marlin |
| *Remora brachyptera* | 13–18 | 4–8 | pelagic | swordfish, marlin, sharks |

All samples were left in 100% EtOH overnight. The following day specimens were critically point dried, sputter coated and visualized with a JCB 2000 SEM microscope (Jeol USA, Inc.).

## 2.5. Histology

Three individual remoras ranging in size from 22.8 to 29.7 cm total length were used for histology, each with eight samples from the lip that were embedded for cross-sectional, frontal and parasagittal sectioning ($n_{total} = 24$). Sample locations were adjacent to those taken for SEM allowing us to make direct comparisons between the cellular morphology and the epithelial topography. The lip surrounding the remora adhesion disc was carefully dissected from individual specimens and fixed in 10% buffered formalin. The lip was then transferred through an ethanol dehydration series until 70% where it was stored. Samples were infiltrated with Electron Microscopy Science JB-4 Embedding media protocol. Each sample was sectioned with either a glass or diamond knife. Sections were transferred onto a glass slide and stained between 1 and 2.5 µm with a modified Lee's Basic Fuchsin and Methylene Blue, which is a general stain that allowed us to identify individual cell types based on cellular morphology. Slides were dipped for 10 s in the working solution (36.0 ml Methylene Blue, 36.0 ml Basic Fuchsin, 63.0 ml phosphate buffer, 45.0 ml 95% EtOH) followed by several quick rinses in dH$_2$O and 95% EtOH to remove background stain.

# 3. Results

Putative mechanoreceptor complexes are only found along the dorsal surface of the fleshy lip of the remora adhesive disc. There is a greater density of mechanoreceptor complexes at the anterior end of the lip than the posterior end (figure 1; $p < 0.001$) for all species. Finally, these putative mechanoreceptor complexes are not grouped together but remain as scattered elements along the entirety of the lip. Putative mechanoreceptor complexes are identified in each of the eight known species of remoras (table 1) by morphological similarity and there was no notable variation in morphology among species. The number of putative mechanoreceptor complexes varied among species, but it is not known if this is owing to interspecific differences or low sample availability.

## 3.1. Epithelial surface morphology

Each putative mechanoreceptor complex is rod-shaped, appearing as a dome-like protrusion from the epithelium under SEM (figure 2). Each mechanoreceptor complex ranges from 100 to 160 µm in diameter and between 50 and 75 µm in height. Two small pores are found adjacent to the centre rod of the complex, probably the artefact of the supportive cartilage found around and beneath the mechanoreceptor complexes. Lack of hair cells extending from the domes or adjacent pores eliminate the possibility of them being neuromasts. Papillae are found uniformly along the lip around the entire circumference of the adhesion disc, this is in contrast to the scattered arrangement of mechanoreceptor

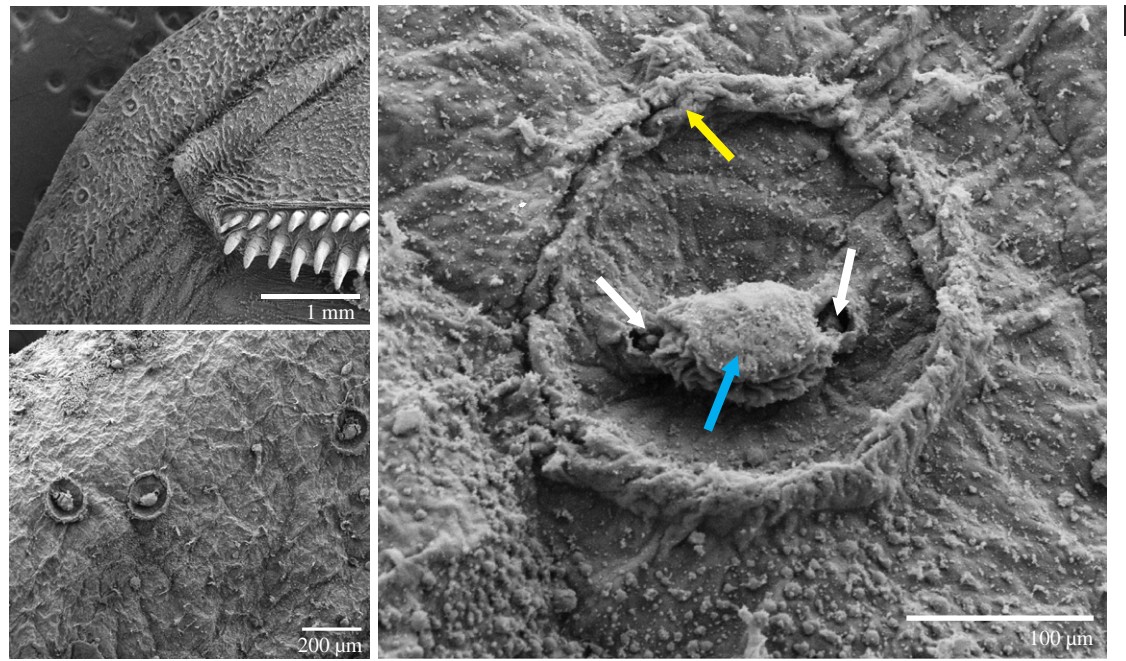

**Figure 2.** Epithelial topography of a putative push-rod receptor. SEM displaying the epithelial topography of the push-rod receptors found along anterior and posterior sections of the lip. Blue arrow: central rod of the cellular complex, white arrows: lateral pores, yellow arrow: peripheral edge of dome.

complexes. Papillae are substantially smaller (approx. 20–30 µm in diameter) and less rotund than the mechanoreceptor complexes. In addition to covering the majority of the soft lip (approximate density of 20 mm$^{-2}$), papillae surround the outer edge of each putative mechanoreceptor complex.

## 3.2. Macro-scale morphology of disc lip

Putative mechanoreceptor complexes in the disc lip of PTA-stained tissues were identified based on comparative measurement, location and density relative to those identified in histological sections and SEM (figure 3a). Soft tissues identified as nervous tissue through morphological comparison to histological sections are higher contrast as compared to surrounding soft tissue because of PTA absorption; papillae do not stain as brightly as the mechanoreceptor complexes nor do they have a ring-like structure around the central rod (figure 3a). Putative mechanoreceptor complexes have an average density of 9–12 cm$^{-2}$ in the anterior lip and 2–4 cm$^{-2}$ in the posterior lip. The nerves innervating each putative push-rod mechanoreceptor complex originate from a large bundle of nerves encircling the ventral portion of the lip (figure 3b,c).

## 3.3. Histology

Each putative push-rod mechanoreceptor complex was cylindrical in shape and composed of several vesicle chains, Merkel cells, all supported by cartilage (figure 4). The top of the putative mechanoreceptor complex is a dome protruding from the epidermal layer and is composed of stratified epithelial tissue (figure 4d). Ventral to this layer, three large nerves can be seen: two peripheral vesicle chains of unmyelinated axons and a singular central nerve (figure 4f). Each of these chains continues through the epidermal and dermal layers of the remora lip and synapse with a single dendrite embedded in the ventral surface of the adhesive disc. Supporting the large nerves are pyramid-shaped pieces of cellular cartilage (figure 4d,e, black arrows). This cartilage is seen throughout the dermal layer and supports the length of the mechanoreceptor complex nerves.

Separating the putative mechanoreceptor complex from the epidermal and dermal layers is loose connective tissue. The nerves continue to penetrate through the lip (figure 3b) and are surrounded by various layers of collagen. A more ventral cross-section of the putative push-rod mechanoreceptor complex reveals a cluster of Merkel cells identified by granule clusters and finger-like projections. The entire putative push-rod mechanoreceptor complex is encapsulated and very distinct from either the

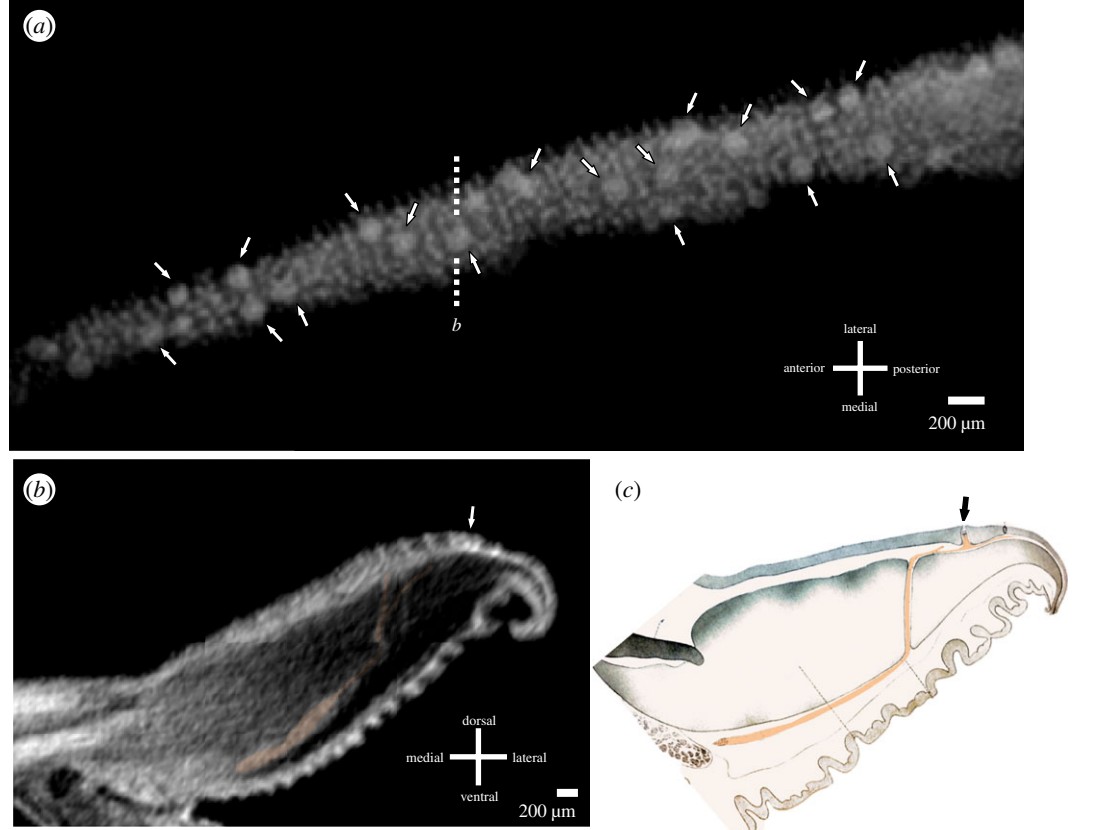

**Figure 3.** Soft tissue of the remora disc lip, visualized by µCT scan with PTA staining. (*a*) Coronal slice through the flat dorsal surface of the distal edge of the anterior remora disc lip. White arrows indicate putative mechanoreceptor complexes amid a field of papillae. The dotted line denotes the location of the cross-section in (*b*). (*b*) Cross-section through anterior remora disc lip with nerve (orange); arrow denotes a putative push-rod mechanoreceptor complex bisected by dotted line in (*a*). (*c*) Illustration from [7] of cross-section through remora disc lip, with putative sensory nerve (orange) and 'sensory sphere' (die Sinneshügel) denoted by the black arrow.

papillae or clusters of free nerve endings in the disc lip. All nerves were presumed to be sensory and not motor, as there is no muscle tissue in the disc lip.

## 4. Discussion

Locomotor and trophic feedback mechanisms rely on physiological pathways informed by nervous tissue. In general, a stimulus acts on either free nerve endings or specialized sensory structures that sense smell, taste, light, pressure waves, temperature, touch, position, chemicals or pain. Sensory transduction of a stimulus carries information along afferent nerve fibres to the brain for environmental processing.

Touch or tactile mechanoreception across vertebrate taxa results from either nonspecific free nerve endings or specific mechanoreceptor complexes [24,30,31]. In fishes, the lateral line system evolved to detect vibration and/or pressure changes in the water column and numerous free nerve endings within the epidermis are presumed to respond to additional tactile cues. Tactile stimuli are important to differentiate suitable habitat or for successful foraging, yet few mechanoreceptors have been described within fishes [31]. Merkel cells are the most commonly observed in some fishes and are known across vertebrates to act as detectors of static stimuli, evoking a slowly adapting train of action potentials upon activation [30]. Among fishes, Merkel cells have been identified in lampreys, hagfish, lungfish, minnows, carp, catfish, and conger eel and are generally found primarily in the mouth, barbels and fins [11,12,16,17,31]. Moray eels use 'touch-corpuscles' within the epithelium of their mouths to discern appropriate prey items [21]. It would appear advantageous to have specific touch receptors in fishes where somatosensory stimuli, the ability to sense pressure, is a crucial behaviour; organisms with such receptors are most commonly tactile foragers.

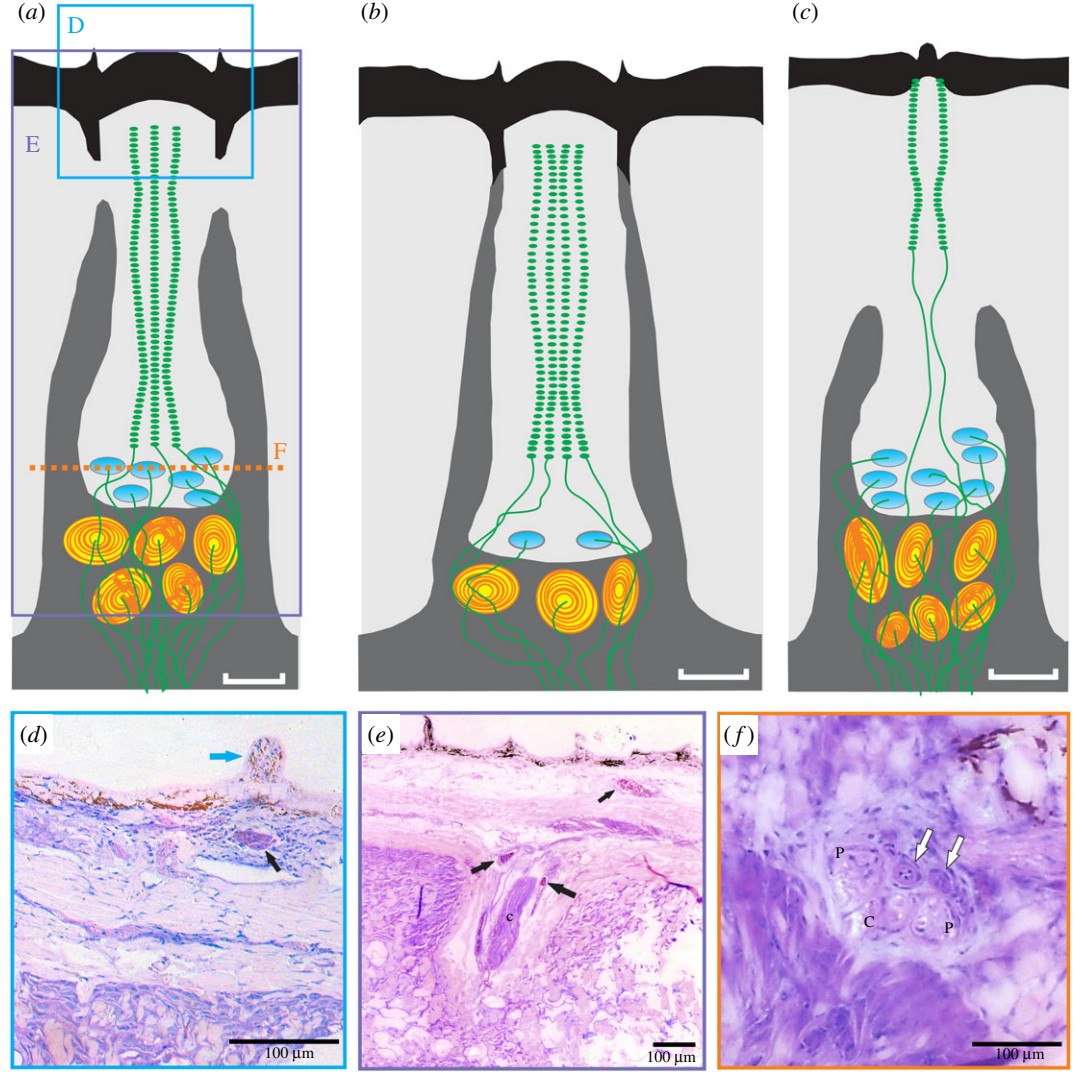

**Figure 4.** Morphology and histology of the putative remora push-rod mechanoreceptor complex. (*a*) Schematic to orient the reader to the histology sections, drawn in the style of the platypus (*b*) and echidna (*c*) push-rod mechanoreceptor complexes as redrawn from [30]. Black, superficial layer of epithelium (keratinized in monotremes); light grey, epidermis; dark grey; dermis surrounding central rod; blue discs, Merkel cells; yellow-orange discs, lamellated (Pacinian) corpuscles; green, afferent nerves. Scale bars are 20 μm. (*d*) Cross-section of a remora push-rod-like mechanoreceptor complex. Blue arrow points to push-rod dome. Black arrow points to cellular support cartilage. (*e*) Cross-section of a remora push-rod mechanoreceptor complex. Black arrows point to cellular cartilage that supports the large central (c) nerve of the cellular complex. The nerve extends through the various connective tissues of the remora lip through the dermis and epidermis layer. (*f*) Coronal section showing the internal morphology of the complex. Two peripheral vesicle chains (P) line the outside of the complex with a central (C) vesicle chain just below the dome. The bottom of the receptor houses a cluster of Merkel cells (white arrows).

## 4.1. Specific tactile mechanoreceptor complexes in vertebrates

A detailed review of cutaneous free nerve endings across vertebrates [30] describes their general abundance and responsiveness to tactile and sensory stimuli; however, specific mechanoreceptor complexes are less widely known across vertebrate taxa. This may be more indicative of a lack of previous research than in their presence or absence in other organisms. Tactile mechanoreceptor complexes are taxon-specific and have been identified in only a few vertebrates. These specialized complexes include tentacles (tentacled snakes), integumentary sensory organs (crocodiles and alligators), Herbst corpuscles (ducks and geese), Eimer's organ (star-nosed moles), push-rods (monotremes) and wing hairs (big-brown bats) [30]. In addition, vibrissae possess Merkel-neurite complexes, are found in all secondarily aquatic mammals, and are known to be used for surface-sensing in pinnipeds, manatees and whales [23–25].

Touch receptor complexes vary morphologically and phylogenetically, having independently evolved in nearly all vertebrate taxa, but the type of push-rod mechanoreceptor complexes described here have only been identified previously in monotremes [26,30,32]. The push-rod receptor complex is a large dome covering a column of cells with vesicle chain receptors, beneath which are several Merkel cell complexes innervated by large myelinated axons [26,30,32]. Vesicle chain receptors are unique to the push-rod complex and are characterized as central and peripheral unmyelinated axons that extend towards the skin surface, have bead-like enlargements at regular intervals, and terminate in expanded bulbous ends. In monotremes, the whole complex is supplied by about 10 myelinated axons, 2 to 3 of which supply vesicle chain receptors [33,34]. Extracellular recordings in monotremes have shown that push-rod mechanoreceptor complexes are slowly adapting with a low threshold for mechanical stimuli and significant dynamic sensitivity [28,35].

## 4.2. Hypothesized mechanosensation in remoras

Embedded in the dermal and epidermal layers of the remora's adhesive disc are putative push-rod mechanoreceptor signalling complexes that in other organisms are known to respond to touch through contact pressure differentials [30,32,34]. These structures in remoras were first observed by Houy [7] who described them as sensory spheres (die Sinneshügel) in his morphological study on the disc but did not attribute a functional role to them. In later works on remora anatomy, these structures were described as possible Meissner corpuscles [6,36], cutaneous sensory structures found in animals where touch is an important sensory modality. Similar to the push-rod receptor complexes found in monotremes, these putative push-rod mechanoreceptor complexes of the remora form a dome pushing outwards under the epidermal layer with three vesicle chains containing the sensory nerves [30,32] and innervated by anterior spinal nerves [6,7]. The whole encapsulated structure is embedded into, but separated from, the epidermal and dermal layers by connective tissue and dermal papillae [37], ensuring that stretching or compression of the skin does not interfere with the nerve response.

Remora adhesion is accomplished by a cascade of hierarchical biomechanical mechanisms, the initiation of which is only possible if the remora is definitively in contact with a host surface. Presumably, the push-rod mechanoreceptor complexes sense contact forces imparted from pressure against a host; attachment contact forces are a combination of suction pressure and elastic yield of the soft tissue of the disc lip [38]. Resting suction pressure differentials (in the absence of shear forces) are on the order of $-0.5 \pm 0.1$ kPa s.e. [8]. However, initial sensing of surface contact may require even lower thresholds of pressure stimulation as attachment to a swimming host organism must occur very rapidly. An additional, and not mutually exclusive, possibility is that some of the putative push-rod mechanoreceptor complexes described herein detect shearing forces relative to the attachment surface and are involved in maintaining adhesion over time. This second function is more likely if the remora's push-rod-like mechanoreceptor complexes are slow-adapting cells with long-term activity profiles as has been shown in monotremes [28,35].

With regards to the shear forces experienced by a remora, it is interesting to note that the estimated average maximum swimming speed for most known marine hosts of remoras is around $3 \text{ m s}^{-1}$, with the exception of only very large hosts, such as the whale shark ($7.3 \text{ m s}^{-1}$), sperm whale ($8.6 \text{ m s}^{-1}$) and blue whale ($10.1 \text{ m s}^{-1}$; [2]). These reported speeds are only size-estimated maximums and it is essential to consider that some hosts may actually swim faster when accounting for muscle physiology and may also swim at the higher end of their speed range more of the time. For example, some billfish are estimated to swim to a maximum speed of $8.3 \text{ m s}^{-1}$, based on muscle contraction time and stride length [39]. Notably, remoras that adhere to pelagic billfish had nearly double the number of mechanoreceptor complexes overall as compared to those species typically associated more with reef fishes. However, the same pattern of more push-rod mechanoreceptor complexes located anteriorly in the disc than posteriorly is true for all remora species regardless of habitat or host association. This suggests that greater sensitivity to mechanical stimuli in the anterior region of the disc is imperative to the function of the remora disc overall, either because attachment is initiated at front of disc [40] or because adhesion loss initiated at the anterior of the disc is more detrimental to maintaining hold.

Here, we describe, to our knowledge, the first evidence of push-rod-like mechanoreceptor complexes in fishes, suggesting that mechanoreception in fishes is more complex than previously thought. Remora adhesion is an undoubtedly unique behaviour, but the physical principles and physiological feedback systems that govern this behaviour are not novel among vertebrates. Sensing the physical attributes of body position relative to the environment is crucial for successful foraging and habitat interaction. Recent studies have shown that the fins of fishes act as proprioceptors [19,20] responding to environmental and

tactile cues that inform fin movement and even deformation from hydrodynamic loading [18,41–43]. Push-rod mechanoreception uses pressure differentials to respond to touch and is probably ideal for organisms that are rheophilic or have adapted a terrestrial-like locomotion that are required to adapt and adjust to changing substrates. Mudskippers, rock climbing gobies, clingfishes, and loaches probably require a pressure-derived feedback system to successfully climb and walk along tough substrates or stay adhered to substrates in the face of fast-moving, turbulent water. The diverse life histories of fishes create an excellent system to further explore the evolution of mechanoreception and proprioception. The appearance of putative push-rod mechanoreceptor complexes in the remora will hopefully encourage researchers to look beyond the lateral line and free nerve endings as primary sensory modalities in fishes and reveal more specific receptors that may explain how fishes fully sense their environments.

Ethics. All study animals were handled humanely and ethically following New Jersey Institute of Technology/Rutgers University IACUC protocol 17058-A0-R1.

Data accessibility. Dryad digital repository link: https://dx.doi.org/10.5061/dryad.t9d744k [44].

Authors' contributions. K.E.C. participated in the design of the study, carried out the histological work, participated in data analysis, and helped draft and revise the manuscript; B.E.F. conceived of the study, participated in the design of the study, coordinated the study, participated in data analysis, prepared the figures, and helped draft and revise the manuscript; C.H.C. conducted the μCT scanning and helped revise the manuscript; L.P.H. participated in the design of the study, assisted with histological work, participated in data analysis and helped edit the manuscript. All authors gave final approval for publication.

Competing interests. The authors declare no financial or non-financial competing interests. At the time of consideration of this manuscript, Prof. Brooke Flammang was a member of the Royal Society Open Science editorial board but had no involvement in the review or assessment of the paper.

Funding. This work was possible thanks to funding from a FY18 faculty seed grant award from NJIT to B.E.F.

Acknowledgements. Special thanks to Flammang laboratory members for fish care, Adam P. Summers and the Karel F. Liem Bioimaging Center for access to imaging and sectioning facilities, funding for microtomes and section materials from the Seaver Institute to Adam P. Summers, and to Kayla Hall for making the microscope camera work at a clutch moment. Dr Ian Malcom, University of Texas Austin, provided helpful commentary.

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
