## [Reviewer comments · Royal Society Open Science]

Review History

RSOS-190990.R0 (Original submission)

Review form: Reviewer 1

Is the manuscript scientifically sound in its present form?

Yes

Are the interpretations and conclusions justified by the results?

No

Is the language acceptable?

Yes

Do you have any ethical concerns with this paper?

No

Have you any concerns about statistical analyses in this paper?

Yes

Recommendation?

Major revision is needed (please make suggestions in comments)

Comments to the Author(s)

The authors use histology, ct scanning, and SEM to document the presence of push rod mechanoreceptors in remoras. The morphology presented is amazing! However, there is only on one species with a small sample size (with a narrow range of animal sizes), and there was no quantification of any structures that were noted as important in the paper. I think adding this would really make this paper more impactful because it would add stronger evidence for the mechanism and importance of these structures in adhesion process (sensing you are stuck.) Most of my specific comments are in regards to trying to quantify something from your existing data set to make this story more substantial. It is really interesting and I am left wanting more from these data. I am excited to see future experiments and work from this team taking into account remora size, different species, etc.

Specific comments:

Line 76: This sentence is a difficult to read and it assumes that something is important for remoras, and this something hasn't really been quantified or measured. At best this is a hypothesis, but not one tested in this paper. It seems like to make this statement true you would need to do ablation experiments to see if pressure sensors are imperative for attachment.

Line 89: This sentence is also difficult to read. 'touch receptor was identified by mammals searching for food'?

Line 94: Are the receptors themselves novel or is it novel that they were observed in remoras / fish?

Line 98: Here is the hypothesis that is stated as truth in line 76.

Line 148: 'A majority of receptors' – can these be quantified in some density or something using a t test among regions you compare? The diagram in Figure 1 is lovely, but it would be great to include some quantitative morphological data. This may actually be difficult to do with such a small sample size. In line 49 it even states that there is a greater density, but I am concerned that this does not seem like a quantitative statement. An idea, can you do figure 1 as a heat map showing density on average around the structure? That would make it more quantitative and give you more with work with in your discussion.

Line 162: Again the statement of papilla being found all over the surface could easily be quantified by dividing the surface into anterior and posterior sections like with the receptors. Is there a way to quantify receptor and papilla ratios. Maybe this is important for understanding the mechanism?

Line 174: Super excited to see this quantified, but earlier you write that it seems to be different in the various regions. This sort gets more interesting if you can actually start to quantify some of the statements made in this paper.

Review form: Reviewer 2

Is the manuscript scientifically sound in its present form?

No

Are the interpretations and conclusions justified by the results?

No

Is the language acceptable?

Yes

Do you have any ethical concerns with this paper?

No

Have you any concerns about statistical analyses in this paper?

No

Recommendation?

Major revision is needed (please make suggestions in comments)

Comments to the Author(s)

This paper describes a newly described structure in the soft tissue on the edge of the remora sucker disk, which is considered by the authors to be push-rod receptors, like those in monotreme mammals. The presence of such structures and especially the possibility that they are like those in mammals, make the implications of the findings in this paper interesting and important to the field of fish and vertebrate biology, especially given the enigmatic nature of remoras, and the fact that the morphology of the sucker disc and its development from modified dorsal fin elements have been described in detail (in prior studies; including one by one of the authors).

However, despite the morphological account provided, some very general statements are provided without citations, some assumptions are made re: function, and thus, the argument that the described structures ARE push-rod receptors is not as compelling as it might be. The paper should really present morphological description with the hypothesis that these structures are push-rod receptors (e.g., putative push-rod receptors, pending more data on innervation and function in vivo), and should more thoroughly discuss the evidence that would support the hypothesis in the Discussion. Further, the writing is quite rough in areas with complex and awkward sentences that need to be re-examined (see line references below).

Given these concerns, I feel that the study is a preliminary one. The manuscript would benefit from addition of and consideration of the following:

1) A more detailed account of what is known about touch receptors in fish, and in other vertebrates, especially monotremes. Perhaps a series of diagrams that provide the same detail as in the last figure of the paper would provide the reader with a better sense of how these newly described structures are similar to touch receptors in other vertebrates. For instance, in addition to whales, (and adult river dolphins, which also possess vibrissae, see Schwerdtfeger et al 1984) all mammals studied that have secondarily returned to aquatic environments possess modified, specialized vibrissae in which there have been extensive characterization of mechanoreceptors (some unique to aquatic taxa) and modification of vibrissal anatomy from their terrestrial counterparts. Some of these papers should likely be cited since vibrissae possess Merkel-neurite complexes as well, which may have more interesting consequences since vibrissae are also used for surface-sensing in both pinnipeds and manatees (see Sarko et. al 2007, Dehnhardt 1994, Marshall et. al 2014; Reep et. al 2001).

2) How are the functional attributes of such a structure defined by its anatomical features? If Merkel cells are found in push-rod receptor complexes (in monotremes), is their presence

diagnostic for such a receptor complex (e.g., where else are Merkel cells found)? How did other studies determine the structure-function relationship in similar structures? The answers to these questions could be a roadmap that could be used to provide a compelling argument for the function of these structures in the remora examined. More sections and other images are needed, especially since the first histological section does not show the entire complex extending to the surface of the skin (e.g., the dome). That being said, descriptions of the putative push-rod receptor complex (and it is a “complex”) could benefit from anatomical analysis using techniques, such as those in papers that have described mechanoreceptors in more detail (the suite of F.L. Rice papers are a good reference), which would also include descriptions of innervation (and CNS projections – to the spinal cord, given that the sucker disk is a derivative of the dorsal fin) using immunohistochemistry, TEM, etc.

3) The Methods section needs to be edited so that the fate of each set of specimens, the tissue samples taken from them [not “sections of the lip”], and the sections of tissue mounted on slides are described clearly. The use of frozen tissue is mentioned in line 113/114, and this is of concern because while this may have been a function of specimen availability, freezing damages tissue, rendering it unsuitable for histology. If live animals are maintained in the lab, why was frozen tissue used at all? Sectioning plane needs to be defined for the tissue in the sucker disk as it relates to the whole animal. The staining protocol needs a citation, and an indication of what cell types are revealed by this staining method (not just “identify individual cell types and cell type morphology”). Finally, both freezing and leaving in 100% ethanol overnight likely introduce shrinkage artifact, but this is not mentioned at all and needs to be addressed.

4) The Results should not name the structures as Push-rod receptors (yet). More objective language is needed – just describe the structures observed. The discussion needs to be significantly reorganized to provide a compelling argument that the structures observed ARE push-rod receptors and how they are similar to and different from those found in monotremes. Such argumentation would require many more citations, which are currently not provided. I would suggest going from more general concepts (e.g., how do known touch receptors work in general, the case in monotremes, and the evidence that allows an argument that the structures observed are quite similar in structure (and thus in function), to the structures in monotremes. This can then be followed by a discussion of how they might function in remoras.

5) Anatomical terms must be used consistently in the text and in the figure captions and labels. Further, they should be used advisedly, if indeed the functional similarity to receptors in monotremes is being asserted. So, labeling the Merkel cells needs some citation backup, as well as “Push-rod dome” in Figure 4, for example.

6) The reference to Augee et al. (which is not in the Lit Cit) is a bit confusing. If Figure 4A is modified from that paper, then what is novel in the current manuscript? What is depicted in the figure used from Augee et al.? What animal did it come from? The caption needs much more detail. AND – in Figure 4, a citation to a paper (4) in German is provided for C. This is a generalized cross-section of the lip – if putative receptor complexes can be seen, label them, and indicate if this paper (presumably read in the original German, if cited) mentioned the receptors.

More specific comments, most of which speak to the issues above:

L ?? What is the common name of the experimental species. Please add this to the Introduction and/or first part of the methods.

L 42/3: are the functional advantages of the sucker disk known or experimentally verified for any remora species?

L 46: I would say “push rod-like receptor complexes”

L 47: “known to”??? Really?

L 60 – different remora species associate with different species of megafauna – this should be mentioned here. What is the preferred association of the species being looked at here? Given the answer, is this species of remora “representative” of the remoras more generally?

L65: “attached under high shear conditions.” – please define shear and add citation.

L 66 – “pectinated” – please define and add citations.

L 73 – “For decades” makes it seem like there has been a decade’s old research effort, but there are only a few papers. Better to say “Over the past ___ decades a small number of studies have....”.

L 74 – thought to be absent....

L 75 – Rohon-Beard cells are transient; what is their function in embryonic fish? Please elaborate on the relevance of these cells to the argument in the manuscript.

L 76 – the definition of the lateral line system (not “canals”) is not articulated correctly. There is a head component and a trunk component, and both sense water flows (which may cause pressure differences sensed by neuromasts) and vibration. See recent references for accuracy of description.

L 77-79 – fix sentence structure – very awkward.

L 80 – “fish skin” – these references are about particular taxa (including non-teleosts, which may be very different from remora), and thus there should be more detail here.

L 85 and throughout – “Moray” is not capitalized.

L 88 – best not to start a paragraph with “While”. Say “Specialized.....in fishes, but they....”

L 89 – You are probably trying to be taxonomically precise, but please define lissamphibians since it is not commonly seen in the non-taxonomic literature.

L 90 – “Infant” – what is more appropriate term for non-humans?

L 92-94 – fix sentence structure – very awkward.

L 94 etc. – THE push-rod receptor – structures need to be defined precisely with citations and perhaps a figure. What are “vesicle chain receptors”? Are they neurons? Are they receptor cells? What is “directionally sensitive feedback? Using the word feedback suggests a neuronal pathway, which is not defined at all.

L 100 – use “newly discovered structures in the epithelium”, and leave assertion of identity as push-rod receptors till the Discussion.

L 130 – what were the CT parameters used (parameters are mentioned in line 133 for other specimens).

L 141 – “additional sections” should be “additional samples”.

L 163 – A putative receptor complex or organ is being described, not just a receptor. Please make this clear throughout.

L 164 – There is no evidence that this is similar to a neuromast. The surface of the skin is not clean, so the absence of apical ciliary bundles found in neuromasts cannot be determined. A papilla is just a “finger” of tissue – so why make a comparison to an “engorged papilla” (and why engorged?

L 166 – not clear what the pores are and where they can be seen in figures.

L 169 – Please start with “In addition to the ____, papillae are found....” (Note – Papillae is the plural). Also, what is meant by “across the entire lip”? Aren’t the found along the lip around the entire circumference of the sucker disk? How small are they? Please give measurements. Are they really papillae (non-descript) or might they be microvilli (supported by actin, for instance)?

L 178 – If histological results are being referenced here, then why not put the histology section first, before this section? What is the best flow for the results to tell a compelling story?

L 181 – “ring like structure” – not clear what this is or where it is illustrated.

L182/183 – There is no evidence in the few images provided that support this description. What is a “circumferential plexus”? Where does this term originate?

L 187/188 – “nerve chains” – are these the “vesicle chains”? Are the “vesicle chains” axons?

L 194 – “receptor nerves” – not well illustrated in the two sections provided. How do you know if they are sensory or motor? Then in L 196 – nerves penetrate through the lip....I don’t see this in

any of the figures. This is where multiple sections, including a 3-D reconstruction would be helpful.

L 198 – This sentence needs to be re-written. I believe it's the receptor complex that is encapsulated, but what "other mechanoreceptors" are being referred to here?

L 222 – start a new paragraph here.

L 223 – make sure what is known has species and citations mentioned here.

L 235 – This section on function deserves its own section in the Discussion.

L 250 – This is repetitive...See larger comment about the need for re-organization of the Discussion above.

Figure 1 – Please give fish size. Distribution of "putative push-rod receptor complexes" should be used. Layers in the enlarged diagram (inset) are confusing. This diagram and labels (!) should include papillae next to the "complex", basement membrane (separating epidermis and dermis), and the two greys are too similar to be seen easily.

Figure 2 – A lower magnification image should be added for context. Why differentiate between anterior and posterior lip regions? (This was not mentioned in the text, I believe). These are not very good SEM's. The skin does not appear to be clean (are there microridges on this skin?) and the surface structure of the surrounding skin cells appears to be distorted or obscured by "goop".

Figure 3 – Nice figure, but first sentence should be "visualized by μ CT with phosphotungstic acid (PTA) staining." Can't see the "dotted line". "Mechanoreceptors" cannot be seen, only the putative "complexes". What is a transaxial slice and how does it differ from a cross-section through the lip?

Figure 4 – Title of figure needs to be more inclusive or the diagrammatic representation should be split into another figure. In addition, one section in each plane is not sufficient to illustrate the structure, which is presumably represented in the diagram.

Decision letter (RSOS-190990.R0)

08-Aug-2019

Dear Professor Flammang,

The editors assigned to your paper ("Knowing when to stick: touch receptors found in the remora adhesive disc") have now received comments from reviewers. We would like you to revise your paper in accordance with the referee and Associate Editor suggestions which can be found below (not including confidential reports to the Editor). Please note this decision does not guarantee eventual acceptance.

Please submit a copy of your revised paper before 31-Aug-2019. Please note that the revision deadline will expire at 00.00am on this date. If we do not hear from you within this time then it will be assumed that the paper has been withdrawn. In exceptional circumstances, extensions may be possible if agreed with the Editorial Office in advance. We do not allow multiple rounds of revision so we urge you to make every effort to fully address all of the comments at this stage. If deemed necessary by the Editors, your manuscript will be sent back to one or more of the original reviewers for assessment. If the original reviewers are not available, we may invite new reviewers.

To revise your manuscript, log into <http://mc.manuscriptcentral.com/rsos> and enter your

Author Centre, where you will find your manuscript title listed under "Manuscripts with Decisions." Under "Actions," click on "Create a Revision." Your manuscript number has been appended to denote a revision. Revise your manuscript and upload a new version through your Author Centre.

- Data accessibility

If you wish to submit your supporting data or code to Dryad (<http://datadryad.org/>), or modify your current submission to dryad, please use the following link:
<http://datadryad.org/submit?journalID=RSOS&manu=RSOS-190990>

- Competing interests

- Authors' contributions

AB carried out the molecular lab work, participated in data analysis, carried out sequence alignments, participated in the design of the study and drafted the manuscript; CD carried out the statistical analyses; EF collected field data; GH conceived of the study, designed the study,

coordinated the study and helped draft the manuscript. All authors gave final approval for publication.

- Acknowledgements

- Funding statement

Kind regards,

on behalf of Dr Jake Socha (Associate Editor) and Kevin Padian (Subject Editor)
openscience@royalsociety.org

Associate Editor's comments (Dr Jake Socha):

The reviewers agree that the morphology described in this manuscript is quite interesting, and the idea that sensors exist that respond to touch and shear in fishes is exciting. However, there are a large number of critiques that need to be addressed in order for this manuscript to be considered for publication. These include a need for more quantification, and bolstering/editing the text to provide greater depth of explanation. It appears that this manuscript was originally written in a short form; I would recommend that the authors take advantage of this journal's more open format to flesh out ideas in greater detail.

Subject Editor's comments (Professor Kevin Padian):

I support the recommendations of the AE and reviewers, and wish you the best success in your revisions. We look forward to the next version.

Reviewers' Comments to Author:

Reviewer: 1

Comments to the Author(s)

The authors use histology, ct scanning, and SEM to document the presence of push rod mechanoreceptors in remoras. The morphology presented is amazing! However, there is only on one species with a small sample size (with a narrow range of animal sizes), and there was no quantification of any structures that were noted as important in the paper. I think adding this would really make this paper more impactful because it would add stronger evidence for the mechanism and importance of these structures in adhesion process (sensing you are stuck.) Most of my specific comments are in regards to trying to quantify something from your existing data

set to make this story more substantial. It is really interesting and I am left wanting more from these data. I am excited to see future experiments and work from this team taking into account remora size, different species, etc.

Specific comments:

Line 76: This sentence is a difficult to read and it assumes that something is important for remoras, and this something hasn't really been quantified or measured. At best this is a hypothesis, but not one tested in this paper. It seems like to make this statement true you would need to do ablation experiments to see if pressure sensors are imperative for attachment.

Line 89: This sentence is also difficult to read. 'touch receptor was identified by mammals searching for food'?

Line 94: Are the receptors themselves novel or is it novel that they were observed in remoras / fish?

Line 98: Here is the hypothesis that is stated as truth in line 76.

Line 148: 'A majority of receptors' – can these be quantified in some density or something using a t test among regions you compare? The diagram in Figure 1 is lovely, but it would be great to include some quantitative morphological data. This may actually be difficult to do with such a small sample size. In line 49 it even states that there is a greater density, but I am concerned that this does not seem like a quantitative statement. An idea, can you do figure 1 as a heat map showing density on average around the structure? That would make it more quantitative and give you more with work with in your discussion.

Line 162: Again the statement of papilla being found all over the surface could easily be quantified by dividing the surface into anterior and posterior sections like with the receptors. Is there a way to quantify receptor and papilla ratios. Maybe this is important for understanding the mechanism?

Line 174: Super excited to see this quantified, but earlier you write that it seems to be different in the various regions. This sort gets more interesting if you can actually start to quantify some of the statements made in this paper.

Reviewer: 2

Comments to the Author(s)

This paper describes a newly described structure in the soft tissue on the edge of the remora sucker disk, which is considered by the authors to be push-rod receptors, like those in monotreme mammals. The presence of such structures and especially the possibility that they are like those in mammals, make the implications of the findings in this paper interesting and important to the field of fish and vertebrate biology, especially given the enigmatic nature of remoras, and the fact that the morphology of the sucker disc and its development from modified dorsal fin elements have been described in detail (in prior studies; including one by one of the authors).

However, despite the morphological account provided, some very general statements are provided without citations, some assumptions are made re: function, and thus, the argument that the described structures ARE push-rod receptors is not as compelling as it might be. The paper should really present morphological description with the hypothesis that these structures are push-rod receptors (e.g., putative push-rod receptors, pending more data on innervation and

function *in vivo*), and should more thoroughly discuss the evidence that would support the hypothesis in the Discussion. Further, the writing is quite rough in areas with complex and awkward sentences that need to be re-examined (see line references below).

Given these concerns, I feel that the study is a preliminary one. The manuscript would benefit from addition of and consideration of the following:

1) A more detailed account of what is known about touch receptors in fish, and in other vertebrates, especially monotremes. Perhaps a series of diagrams that provide the same detail as in the last figure of the paper would provide the reader with a better sense of how these newly described structures are similar to touch receptors in other vertebrates. For instance, in addition to whales, (and adult river dolphins, which also possess vibrissae, see Schwerdtfeger et al 1984) all mammals studied that have secondarily returned to aquatic environments possess modified, specialized vibrissae in which there have been extensive characterization of mechanoreceptors (some unique to aquatic taxa) and modification of vibrissal anatomy from their terrestrial counterparts. Some of these papers should likely be cited since vibrissae possess Merkel-neurite complexes as well, which may have more interesting consequences since vibrissae are also used for surface-sensing in both pinnipeds and manatees (see Sarko et. al 2007, Dehnhardt 1994, Marshall et. al 2014; Reep et. al 2001).

2) How are the functional attributes of such a structure defined by its anatomical features? If Merkel cells are found in push-rod receptor complexes (in monotremes), is their presence diagnostic for such a receptor complex (e.g., where else are Merkel cells found)? How did other studies determine the structure-function relationship in similar structures? The answers to these questions could be a roadmap that could be used to provide a compelling argument for the function of these structures in the remora examined. More sections and other images are needed, especially since the first histological section does not show the entire complex extending to the surface of the skin (e.g., the dome). That being said, descriptions of the putative push-rod receptor complex (and it is a “complex”) could benefit from anatomical analysis using techniques, such as those in papers that have described mechanoreceptors in more detail (the suite of F.L. Rice papers are a good reference), which would also include descriptions of innervation (and CNS projections - to the spinal cord, given that the sucker disk is a derivative of the dorsal fin) using immunohistochemistry, TEM, etc.

3) The Methods section needs to be edited so that the fate of each set of specimens, the tissue samples taken from them [not “sections of the lip”], and the sections of tissue mounted on slides are described clearly. The use of frozen tissue is mentioned in line 113/114, and this is of concern because while this may have been a function of specimen availability, freezing damages tissue, rendering it unsuitable for histology. If live animals are maintained in the lab, why was frozen tissue used at all? Sectioning plane needs to be defined for the tissue in the sucker disk as it relates to the whole animal. The staining protocol needs a citation, and an indication of what cell types are revealed by this staining method (not just “identify individual cell types and cell type morphology”). Finally, both freezing and leaving in 100% ethanol overnight likely introduce shrinkage artifact, but this is not mentioned at all and needs to be addressed.

4) The Results should not name the structures as Push-rod receptors (yet). More objective language is needed - just describe the structures observed. The discussion needs to be significantly reorganized to provide a compelling argument that the structures observed ARE push-rod receptors and how they are similar to and different from those found in monotremes. Such argumentation would require many more citations, which are currently not provided. I would suggest going from more general concepts (e.g., how do known touch receptors work in general, the case in monotremes, and the evidence that allows an argument that the structures

observed are quite similar in structure (and thus in function), to the structures in monotremes. This can then be followed by a discussion of how they might function in remoras.

5) Anatomical terms must be used consistently in the text and in the figure captions and labels. Further, they should be used advisedly, if indeed the functional similarity to receptors in monotremes is being asserted. So, labeling the Merkel cells needs some citation backup, as well as “Push-rod dome” in Figure 4, for example.

6) The reference to Augee et al. (which is not in the Lit Cit) is a bit confusing. If Figure 4A is modified from that paper, then what is novel in the current manuscript? What is depicted in the figure used from Augee et al.? What animal did it come from? The caption needs much more detail. AND – in Figure 4, a citation to a paper (4) in German is provided for C. This is a generalized cross-section of the lip – if putative receptor complexes can be seen, label them, and indicate if this paper (presumably read in the original German, if cited) mentioned the receptors.

More specific comments, most of which speak to the issues above:

L ?? What is the common name of the experimental species. Please add this to the Introduction and/or first part of the methods.

L 42/3: are the functional advantages of the sucker disk known or experimentally verified for any remora species?

L 46: I would say “push rod-like receptor complexes”

L 47: “known to”??? Really?

L 60 – different remora species associate with different species of megafauna – this should be mentioned here. What is the preferred association of the species being looked at here? Given the answer, is this species of remora “representative” of the remoras more generally?

L65: “attached under high shear conditions.” – please define shear and add citation.

L 66 – “pectinated” – please define and add citations.

L 73 – “For decades” makes it seem like there has been a decade’s old research effort, but there are only a few papers. Better to say “Over the past __ decades a small number of studies have....”.

L 74 – thought to be absent....

L 75 – Rohon-Beard cells are transient; what is their function in embryonic fish? Please elaborate on the relevance of these cells to the argument in the manuscript.

L 76 – the definition of the lateral line system (not “canals”) is not articulated correctly. There is a head component and a trunk component, and both sense water flows (which may cause pressure differences sensed by neuromasts) and vibration. See recent references for accuracy of description.

L 77-79 – fix sentence structure – very awkward.

L 80 – “fish skin” – these references are about particular taxa (including non-teleosts, which may be very different from remora), and thus there should be more detail here.

L 85 and throughout – “Moray” is not capitalized.

L 88 – best not to start a paragraph with “While”. Say “Specialized.....in fishes, but they....”

L 89 – You are probably trying to be taxonomically precise, but please define lissamphibians since it is not commonly seen in the non-taxonomic literature.

L 90 – “Infant” – what is more appropriate term for non-humans?

L 92-94 – fix sentence structure – very awkward.

L 94 etc. – THE push-rod receptor – structures need to be defined precisely with citations and perhaps a figure. What are “vesicle chain receptors”? Are they neurons? Are they receptor cells? What is “directionally sensitive feedback? Using the word feedback suggests a neuronal pathway, which is not defined at all.

L 100 – use “newly discovered structures in the epithelium”, and leave assertion of identity as push-rod receptors till the Discussion.

L 130 – what were the CT parameters used (parameters are mentioned in line 133 for other specimens).

L 141 – “additional sections” should be “additional samples”.

L 163 – A putative receptor complex or organ is being described, not just a receptor. Please make this clear throughout.

L 164 – There is no evidence that this is similar to a neuromast. The surface of the skin is not clean, so the absence of apical ciliary bundles found in neuromasts cannot be determined. A papilla is just a “finger” of tissue – so why make a comparison to an “engorged papilla” (and why engorged)?

L 166 – not clear what the pores are and where they can be seen in figures.

L 169 – Please start with “In addition to the ____, papillae are found....” (Note – Papillae is the plural). Also, what is meant by “across the entire lip”? Aren’t the found along the lip around the entire circumference of the sucker disk? How small are they? Please give measurements. Are they really papillae (non-descript) or might they be microvilli (supported by actin, for instance)?

L 178 – If histological results are being referenced here, then why not put the histology section first, before this section? What is the best flow for the results to tell a compelling story?

L 181 – “ring like structure” – not clear what this is or where it is illustrated.

L182/183 – There is no evidence in the few images provided that support this description. What is a “circumferential plexus”? Where does this term originate?

L 187/188 – “nerve chains” – are these the “vesicle chains”? Are the “vesicle chains” axons?

L 194 – “receptor nerves” – not well illustrated in the two sections provided. How do you know if they are sensory or motor? Then in L 196 – nerves penetrate through the lip...I don’t see this in any of the figures. This is where multiple sections, including a 3-D reconstruction would be helpful.

L 198 – This sentence needs to be re-written. I believe it’s the receptor complex that is encapsulated, but what “other mechanoreceptors” are being referred to here?

L 222 – start a new paragraph here.

L 223 – make sure what is known has species and citations mentioned here.

L 235 – This section on function deserves its own section in the Discussion.

L 250 – This is repetitive...See larger comment about the need for re-organization of the Discussion above.

Figure 1 – Please give fish size. Distribution of “putative push-rod receptor complexes” should be used. Layers in the enlarged diagram (inset) are confusing. This diagram and labels (!) should include papillae next to the “complex”, basement membrane (separating epidermis and dermis), and the two greys are too similar to be seen easily.

Figure 2 – A lower magnification image should be added for context. Why differentiate between anterior and posterior lip regions? (This was not mentioned in the text, I believe). These are not very good SEM’s. The skin does not appear to be clean (are there microridges on this skin?) and the surface structure of the surrounding skin cells appears to be distorted or obscured by “goop”.

Figure 3 – Nice figure, but first sentence should be “visualized by μ CT with phosphotungstic acid (PTA) staining.” Can’t see the “dotted line”. “Mechanoreceptors” cannot be seen, only the putative “complexes”. What is a transaxial slice and how does it differ from a cross-section through the lip?

Figure 4 – Title of figure needs to be more inclusive or the diagrammatic representation should be split into another figure. In addition, one section in each plane is not sufficient to illustrate the structure, which is presumably represented in the diagram.

Author's Response to Decision Letter for (RSOS-190990.R0)

See Appendix A.

RSOS-190990.R1 (Revision)

Review form: Reviewer 1

Is the manuscript scientifically sound in its present form?

Yes

Are the interpretations and conclusions justified by the results?

Yes

Is the language acceptable?

Yes

Do you have any ethical concerns with this paper?

No

Have you any concerns about statistical analyses in this paper?

Yes

Recommendation?

Accept as is

Comments to the Author(s)

This revisions is much improved and the authors added great information to the morphology aspects (gorgeous figures!), but the quantification is of these receptors is still lacking. There is one t-test on $n=3$; without the quantification this story seems superficial. Understanding the arrangement and densities might help infer function or mechanisms.

Decision letter (RSOS-190990.R1)

22-Nov-2019

Dear Professor Flammang:

On behalf of the Editors, I am pleased to inform you that your Manuscript RSOS-190990.R1 entitled "Knowing when to stick: touch receptors found in the remora adhesive disc" has been accepted for publication in Royal Society Open Science subject to minor revision in accordance with the referee suggestions. Please find the referees' comments at the end of this email.

The reviewers and Subject Editor have recommended publication, but also suggest some minor

revisions to your manuscript. Therefore, I invite you to respond to the comments and revise your manuscript.

- Ethics statement

- Data accessibility

If you wish to submit your supporting data or code to Dryad (<http://datadryad.org/>), or modify your current submission to dryad, please use the following link:
<http://datadryad.org/submit?journalID=RSOS&manu=RSOS-190990.R1>

- Competing interests

Please declare any financial or non-financial competing interests, or state that you have no competing interests. We also ask that you please add the following sentence to your Competing Interests section:

"At the time of consideration of this manuscript, Professor Brooke Flammang was a member of the Royal Society Open Science editorial board, but had no involvement in the review or assessment of the paper."

- Authors' contributions

- Acknowledgements

- Funding statement

Because the schedule for publication is very tight, it is a condition of publication that you submit the revised version of your manuscript before 01-Dec-2019. Please note that the revision deadline will expire at 00.00am on this date. If you do not think you will be able to meet this date please let me know immediately.

Kind regards,

on behalf of Dr Jake Socha (Associate Editor) and Kevin Padian (Subject Editor)
openscience@royalsociety.org

Associate Editor Comments to Author (Dr Jake Socha):

The authors have satisfyingly addressed the comments and concerns of both reviewers. Although there is still some question about statistical comparisons, the limited access to specimens and the tentative framing of the result (as 'putative') is justification for moving forward to publication. This paper provides the first evidence of push-rod-like mechanoreceptor complexes in fishes, and thus will serve as motivation for future studies to delve into the problem. Congratulations on a nice piece of morphological work.

One small edit to add: in line 45, change the text to "push-rod-like" (as in line 48).

Reviewer comments to Author:

Reviewer: 1

Comments to the Author(s)

This revisions is much improved and the authors added great information to the morphology aspects (gorgeous figures!), but the quantification is of these receptors is still lacking. There is one t-test on $n=3$; without the quantification this story seems superficial. Understanding the arrangement and densities might help infer function or mechanisms.

Author's Response to Decision Letter for (RSOS-190990.R1)

See Appendix B.

Decision letter (RSOS-190990.R2)

05-Dec-2019

Dear Professor Flammang,

It is a pleasure to accept your manuscript entitled "Knowing when to stick: touch receptors found in the remora adhesive disc" in its current form for publication in Royal Society Open Science.

The comments of the reviewer(s) who reviewed your manuscript are included at the foot of this letter.

Best regards,

on behalf of Dr Jake Socha (Associate Editor) and Kevin Padian (Subject Editor)
openscience@royalsociety.org

Associate Editor Comments to Author (Dr Jake Socha):

Congratulations again! Looking forward to seeing what becomes of this discovery.

Appendix A

Dear Editors,

Following is our revised manuscript submission entitled “Knowing when to stick: touch receptors found in the remora adhesive disc” for consideration for publication in Royal Society Open Science. We are extremely appreciative of the thorough and thoughtful insights provided by the two reviewers and Associate Editor Socha and are very grateful for their efforts and contributions. We have followed all their suggested recommendations (please see our replies in ****blue**** below) and hope you find our submission much improved.

We have followed the reviewers suggestions on all counts. In summary, our changes include new data on the putative push-rod mechanoreceptor complexes in all 8 known species of remoras (Table 1), a thorough rewrite of the discussion, and complete overhaul of figures 1, 2, and 4 (including new histological images). We have also taken great care to be clear that these are putative push-rod mechanoreceptor complexes described based on morphological similarity and be less declarative in asserting their function.

Associate Editor's comments (Dr Jake Socha):

The reviewers agree that the morphology described in this manuscript is quite interesting, and the idea that sensors exist that respond to touch and shear in fishes is exciting. However, there are a large number of critiques that need to be addressed in order for this manuscript to be considered for publication. These include a need for more quantification, and bolstering/editing the text to provide greater depth of explanation. It appears that this manuscript was originally written in a short form; I would recommend that the authors take advantage of this journal's more open format to flesh out ideas in greater detail.

Subject Editor's comments (Professor Kevin Padian):

I support the recommendations of the AE and reviewers, and wish you the best success in your revisions. We look forward to the next version.

Reviewers' Comments to Author:

Reviewer: 1

Comments to the Author(s)

The authors use histology, ct scanning, and SEM to document the presence of push rod mechanoreceptors in remoras. The morphology presented is amazing! However, there is only on one species with a small sample size (with a narrow range of animal sizes), and there was no quantification of any structures that were noted as important in the paper. I think adding this would really make this paper more impactful because it would add stronger evidence for the mechanism and importance of these structures in adhesion process (sensing you are stuck.) Most of my specific comments are in regards to trying to quantify something from your existing data set to make this story more substantial. It is really interesting and I am left wanting more from these data. I am excited to see future experiments and work from this team taking into account remora size, different species, etc.

****Thank you, we are glad you enjoyed the morphological presentation! We have added a statement in the methods to clarify our rationale for species choice: namely, it is the only species available from commercial suppliers. Without approaching a host organism and trying to forcibly remove a remora, they are seldom caught, and even more rarely available alive. That being said, we were able to obtain access to museum specimens representing each of the 8 species of remora and have now included a table**

showing the comparative abundances of these mechanoreceptor complexes among them and a paragraph in the discussion hypothesizing what the interspecific variation might affect. Unfortunately the sample size for each species is still low, but that was not possible to reconcile given the short period of time allowed for revisions. Because these are museum specimens, we did not have permission to section them for histological comparison. We have also now performed a t-test to compare the density of mechanoreceptor complexes in the anterior and posterior lip of the disc.

Specific comments:

Line 76: This sentence is a difficult to read and it assumes that something is important for remoras, and this something hasn't really been quantified or measured. At best this is a hypothesis, but not one tested in this paper. It seems like to make this statement true you would need to do ablation experiments to see if pressure sensors are imperative for attachment.

** We have separated this into 2 sentences and made it less declarative: "Specific pressure feedback would seem imperative for remoras to know when they have achieved contact and then initiate attachment. Presumably, pressure feedback would be important to any fishes that physically interact with a solid environment."

Line 89: This sentence is also difficult to read. 'touch receptor was identified by mammals searching for food'?

** That was indeed quite difficult. We have changed it to: "touch receptor has been identified in mammals that habitually search for food"

Line 94: Are the receptors themselves novel or is it novel that they were observed in remoras / fish?

**They are new to fishes, and we have reworded this as: "we describe putative push-rod mechanoreceptor complexes found for the first time in fishes"

Line 98: Here is the hypothesis that is stated as truth in line 76.

**We have corrected the statement in line 76 (above) to sound less presumptuous and more as part of the support building towards our hypothesis here.

Line 148: 'A majority of receptors' – can these be quantified in some density or something using a t test among regions you compare? The diagram in Figure 1 is lovely, but it would be great to include some quantitative morphological data. This may actually be difficult to do with such a small sample size. In line 49 it even states that there is a greater density, but I am concerned that this does not seem like a quantitative statement. An idea, can you do figure 1 as a heat map showing density on average around the structure? That would make it more quantitative and give you more work with in your discussion.

** A t-test was used to compare the abundance of putative push-rod mechanoreceptor complexes in the anterior and posterior regions of the disc (n=3). Fig 1 diagram has been modified to be more accurate in representing the relative abundance of receptor complexes, and 2 new panels have been added to show the difference in anterior vs posterior lip distribution. In addition, we have now surveyed the mechanoreceptor complex distribution among specimens of each species of remora for comparison.

Line 162: Again the statement of papilla being found all over the surface could easily be quantified by dividing the surface into anterior and posterior sections like with the receptors. Is there a way to quantify receptor and papilla ratios. Maybe this is important for understanding the mechanism?

**The papillae are uniformly distributed as has now been stated in text. We have included their

approximate density of 20 mm⁻² as well. We do not believe that the papillae serve a large sensory role here (and are therefore not the focus of this paper), but instead play a functional role in adhesion. To this end they are being discussed in a functional context in a separate paper dealing with adhesive mechanics of the soft lip that is currently in preparation and hopefully submitted for publication very soon after this revision is returned to you.

Line 174: Super excited to see this quantified, but earlier you write that it seems to be different in the various regions. This sort gets more interesting if you can actually start to quantify some of the statements made in this paper.

** Thank you - we have added more quantification as suggested, please see changes made to the above comments and regarding previous lines 148 specifically.

Reviewer: 2

Comments to the Author(s)

This paper describes a newly described structure in the soft tissue on the edge of the remora sucker disk, which is considered by the authors to be push-rod receptors, like those in monotreme mammals. The presence of such structures and especially the possibility that they are like those in mammals, make the implications of the findings in this paper interesting and important to the field of fish and vertebrate biology, especially given the enigmatic nature of remoras, and the fact that the morphology of the sucker disc and its development from modified dorsal fin elements have been described in detail (in prior studies; including one by one of the authors).

However, despite the morphological account provided, some very general statements are provided without citations, some assumptions are made re: function, and thus, the argument that the described structures ARE push-rod receptors is not as compelling as it might be. The paper should really present morphological description with the hypothesis that these structures are push-rod receptors (e.g., putative push-rod receptors, pending more data on innervation and function in vivo), and should more thoroughly discuss the evidence that would support the hypothesis in the Discussion. Further, the writing is quite rough in areas with complex and awkward sentences that need to be re-examined (see line references below).

Given these concerns, I feel that the study is a preliminary one. The manuscript would benefit from addition of and consideration of the following:

1) A more detailed account of what is known about touch receptors in fish, and in other vertebrates, especially monotremes. Perhaps a series of diagrams that provide the same detail as in the last figure of the paper would provide the reader with a better sense of how these newly described structures are similar to touch receptors in other vertebrates. For instance, in addition to whales, (and adult river dolphins, which also possess vibrissae, see Schwerdtfeger et al 1984) all mammals studied that have secondarily returned to aquatic environments possess modified, specialized vibrissae in which there have been extensive characterization of mechanoreceptors (some unique to aquatic taxa) and modification of vibrissal anatomy from their terrestrial counterparts. Some of these papers should likely be cited since vibrissae possess Merkel-neurite complexes as well, which may have more interesting consequences since vibrissae are also used for surface-sensing in both pinnipeds and manatees (see Sarko et. al 2007, Dehnhardt 1994, Marshall et. al 2014; Reep et. al 2001).

** We have revised this section extensively as part of rewriting the discussion. This section (lines 233-275) now includes mention of all known mechanoreception in fishes as well as all specific touch receptor complexes known in vertebrates, and cite several review papers which cover these topics in great detail,

including a comparative diagram like that suggested by the reviewer (Schneider et al 2016). We have also revised Fig 4 substantially: it now includes a new schematic diagram of the remora push-rod mechanoreceptor complex with direct comparison to the push-rod mechanoreceptor complexes of platypus and echidnas (Fig 4B,C) a la Schneider et al (2016). We also performed additional histological sectioning to show the push-rod dome in cross-section (Fig 4D).

2) How are the functional attributes of such a structure defined by its anatomical features? If Merkel cells are found in push-rod receptor complexes (in monotremes), is their presence diagnostic for such a receptor complex (e.g., where else are Merkel cells found)? How did other studies determine the structure-function relationship in similar structures? The answers to these questions could be a roadmap that could be used to provide a compelling argument for the function of these structures in the remora examined. More sections and other images are needed, especially since the first histological section does not show the entire complex extending to the surface of the skin (e.g., the dome). That being said, descriptions of the putative push-rod receptor complex (and it is a “complex”) could benefit from anatomical analysis using techniques, such as those in papers that have described mechanoreceptors in more detail (the suite of F.L. Rice papers are a good reference), which would also include descriptions of innervation (and CNS projections – to the spinal cord, given that the sucker disk is a derivative of the dorsal fin) using immunohistochemistry, TEM, etc.

** We have expanded our discussion of Merkel cells as well as the diagnostic characteristics of push-rods in lines 264-275. The push rod receptor is marked by the presence of Merkel cells but not by itself. It is the rod complex that is the identifying feature. The rod is composed of a central vesicle chain, peripheral vesicle chain, and a cluster of Merkel cells at the base (Manger and Petigrew 1996). The structure-function relationship was determined in monotremes using extracellular recordings, as described in lines 273-275. We have performed additional histological sectioning and were successful in imaging the push-rod dome (Fig 4D). These structures are identifiable by morphology and do not require a tissue specific stain for identification.

3) The Methods section needs to be edited so that the fate of each set of specimens, the tissue samples taken from them [not “sections of the lip”], and the sections of tissue mounted on slides are described clearly. The use of frozen tissue is mentioned in line 113/114, and this is of concern because while this may have been a function of specimen availability, freezing damages tissue, rendering it unsuitable for histology. If live animals are maintained in the lab, why was frozen tissue used at all? Sectioning plane needs to be defined for the tissue in the sucker disk as it relates to the whole animal. The staining protocol needs a citation, and an indication of what cell types are revealed by this staining method (not just “identify individual cell types and cell type morphology”).

**We have revised the Methods section as suggested. With particular consideration of the freezing of tissue, we have added the statement: “Freezing of specimens was necessary to transfer then from NJIT to the University of Washington and George Washington University, where histological and scanning electron microscopy studies were conducted.” We have now added specific details describing the staining protocol thoroughly.

4) The Results should not name the structures as Push-rod receptors (yet). More objective language is needed – just describe the structures observed. The discussion needs to be significantly reorganized to provide a compelling argument that the structures observed ARE push-rod receptors and how they are similar to and different from those found in monotremes. Such argumentation would require many more citations, which are currently not provided. I would suggest going from more general concepts (e.g., how do known touch receptors work in general, the case in monotremes, and the evidence that allows an

argument that the structures observed are quite similar in structure (and thus in function), to the structures in monotremes. This can then be followed by a discussion of how they might function in remoras.

**We have revised the results to refer to the morphology as putative mechanoreceptor complexes and used more objective language as requested. In addition, we have followed your suggestions below and completely rewritten the discussion to provide more evidence for our conclusions, including incorporating more citations.

5) Anatomical terms must be used consistently in the text and in the figure captions and labels. Further, they should be used advisedly, if indeed the functional similarity to receptors in monotremes is being asserted. So, labeling the Merkel cells needs some citation backup, as well as “Push-rod dome” in Figure 4, for example.

**We have made a point of using the full phrase of “putative push-rod mechanoreceptor complexes” throughout and not just “push-rods” or ‘mechanoreceptors” as we had previously in places. The Merkel cells and push-rod dome are referenced when Figure 4 is discussed in the text.

6) The reference to Augée et al. (which is not in the Lit Cit) is a bit confusing. If Figure 4A is modified from that paper, then what is novel in the current manuscript? What is depicted in the figure used from Augée et al.? What animal did it come from? The caption needs much more detail. AND – in Figure 4, a citation to a paper (4) in German is provided for C. This is a generalized cross-section of the lip – if putative receptor complexes can be seen, label them, and indicate if this paper (presumably read in the original German, if cited) mentioned the receptors.

** Please note that Augée was cited (reference 27 in the original version), just missed in the figure legend using the reference software. The Augée paper is on echidnas. As stated previously, push-rod mechanoreceptors were previously only known to monotremes. Our paper is novel because it presents the first time this structure has been identified in fishes, and the first time a specific mechanoreceptor complex has been identified in fishes. We have now included the species information in the figure legend as suggested. NB: the Augée component was removed in the full revision of Fig 4.

We have added the following in the discussion: “The structures were first observed by Houy (1909) who described them as sensory spheres (die Sinneshügel) in his morphological study on the disc but did not attribute a functional role”, as well as now indicating the putative receptors in figure 3C with an arrow.

More specific comments, most of which speak to the issues above:

L ?? What is the common name of the experimental species. Please add this to the Introduction and/or first part of the methods.

** It is commonly known as the sharksucker and this has been added to the methods where the species is first mentioned by name.

L 42/3: are the functional advantages of the sucker disk known or experimentally verified for any remora species?

**This comment is referring to the Abstract, and cannot be referenced here, but the functional advantages are described in the first paragraph of the introduction with references.

L 46: I would say “push rod-like receptor complexes”

**changed as suggested

L 47: “known to”??? Really?

**They are known to respond to touch and shear in monotremes, where they are otherwise described (but perhaps found more widely just not yet looked for in other vertebrates). We have added “in other organisms” to be more accurate.

L 60 – different remora species associate with different species of megafauna – this should be mentioned here. What is the preferred association of the species being looked at here? Given the answer, is this species of remora “representative” of the remoras more generally?

**This is a great point. Actually *Echeneis naucrates*, the species we used, is the most generalist of all the remora species, adhering to almost everything, and is therefore the best representative of the other remoras possible. We have added this text to the methods under “Sample collection” where we indicate species choice rationale.

L65: “attached under high shear conditions.” – please define shear and add citation.

** We have now added: “where drag forces impart pressure strain inducing sliding relative to the host (Beckert et al 2016).”

L 66 – “pectinated” – please define and add citations.

** Pectinated = Having narrow ridges or projections aligned close together like the teeth of a comb. In text we have added “or comb-like” and cited Britz and Johnson (2012) who give a nice description of this morphology and describe it as such.

L 73 – “For decades” makes it seem like there has been a decade’s old research effort, but there are only a few papers. Better to say “Over the past ___ decades a small number of studies have....”.

**Changed exactly as suggested.

L 74 – thought to be absent....

**Corrected.

L 75 – Rohon-Beard cells are transient; what is their function in embryonic fish? Please elaborate on the relevance of these cells to the argument in the manuscript.

** These cells are not relevant in this context and this phrasing was removed.

L 76 – the definition of the lateral line system (not “canals”) is not articulated correctly. There is a head component and a trunk component, and both sense water flows (which may cause pressure differences sensed by neuromasts) and vibration. See recent references for accuracy of description.

** Thank you for this helpful clarification. We have now edited this sentence to read: “fishes sense their environment predominantly through a lateral line canal system on their head and body (8) that is sensitive to water flow and vibrations in the environment (14).”

L 77-79 – fix sentence structure – very awkward.

**It was indeed terribly awkward, as Rev 1 also pointed out, and we have changed it (please see above).

L 80 – “fish skin” – these references are about particular taxa (including non-teleosts, which may be very different from remora), and thus there should be more detail here.

**Change to “epidermis of teleosts as well as hagfish”

L 85 and throughout – “Moray” is not capitalized.

**Fixed!

L 88 – best not to start a paragraph with “While”. Say “Specialized.....in fishes, but they....”

**Changed as suggested.

L 89 – You are probably trying to be taxonomically precise, but please define lissamphibians since it is not commonly seen in the non-taxonomic literature.

**Changed to amphibians, because to our knowledge that is still technically correct here.

L 90 – “Infant” – what is more appropriate term for non-humans?

**Changed to “Toothed whale calves”

L 92-94 - fix sentence structure – very awkward.

**Please see revision in response to Rev 1 above.

L 94 etc. – THE push-rod receptor – structures need to be defined precisely with citations and perhaps a figure. What are “vesicle chain receptors”? Are they neurons? Are they receptor cells? What is “directionally sensitive feedback? Using the word feedback suggests a neuronal pathway, which is not defined at all.

** Vesicle chain receptors are unmyelinated axons as described in lines 269-273 and Manger & Hughes (1992) and Iggo et al (1985). We changed “directionally sensitive feedback” to “anisotropic sensory information”. Figure4 shows the structures that make up the push-rod mechanoreceptor complex.

L 100 – use “newly discovered structures in the epithelium”, and leave assertion of identity as push-rod receptors till the Discussion.

**We have revised this to read: “we describe a putative mechanoreceptor complex found for the first time in fishes, in the epithelium of the remora adhesive disc”

L 130 – what were the CT parameters used (parameters are mentioned in line 133 for other specimens).

** We have now added the settings in text (125 uA, 80 kV, with 47 ms exposure time, with a voxel size of 31 µm).

L 141 – “additional sections” should be “additional samples”.

**Changed as suggested.

L 163 – A putative receptor complex or organ is being described, not just a receptor. Please make this clear throughout.

**We have corrected this phrasing throughout the manuscript.

L 164 – There is no evidence that this is similar to a neuromast. The surface of the skin is not clean, so the absence of apical ciliary bundles found in neuromasts cannot be determined. A papilla is just a “finger” of tissue – so why make a comparison to an “engorged papilla” (and why engorged?)

**This attempt at comparison was deemed not helpful or necessary and was therefore removed.

L 166 – not clear what the pores are and where they can be seen in figures.

** The lateral pores are now marked with white arrows in the revised Figure 2.

L 169 – Please start with “In addition to the _____, papillae are found...” (Note – Papillae is the plural). Also, what is meant by “across the entire lip”? Aren’t they found along the lip around the entire circumference of the sucker disk? How small are they? Please give measurements. Are they really papillae (non-descript) or might they be microvilli (supported by actin, for instance)?

**Thank you for catching our typo! We have followed your suggested phrasing. papillae diameter (approx 20 micron) is now given in text.

L 178 – If histological results are being referenced here, then why not put the histology section first, before this section? What is the best flow for the results to tell a compelling story?

** We have reorganized these sections for better flow as suggested.

L 181 – “ring like structure” – not clear what this is or where it is illustrated.

** The ring structure is the edge of the dome, now marked by the yellow arrow in the new and improved Fig 2.

L182/183 – There is no evidence in the few images provided that support this description. What is a “circumferential plexus”? Where does this term originate?

** This was poorly stated and changed to read: “The nerves innervating each mechanoreceptor originate from a large bundle of nerves encircling the ventral portion of the lip”

L 187/188 – “nerve chains” – are these the “vesicle chains”? Are the “vesicle chains” axons?

** This has been rewritten to read: “Vesicle chain receptors are unique to the push-rod complex and are characterized as central and peripheral unmyelinated axons that extend toward the skin surface, have bead-like enlargements at regular intervals, and terminate in expanded bulbous ends.”.

L 194 – “receptor nerves” – not well illustrated in the two sections provided. How do you know if they are sensory or motor? Then in L 196 – nerves penetrate through the lip....I don't see this in any of the figures. This is where multiple sections, including a 3-D reconstruction would be helpful.

**There are no muscles in the remora lip (line 229-230); therefore they could not be motor. Figure 3B shows the nerve passing through the lip.

L 198 – This sentence needs to be re-written. I believe it's the receptor complex that is encapsulated, but what “other mechanoreceptors” are being referred to here?

**Thank you - we were referring to papillae here and have corrected it to be more clear.

L 222 – start a new paragraph here.

** Done.

L 223 – make sure what is known has species and citations mentioned here.

** Please see lines 259-261: Specific tactile mechanoreceptor complexes have been identified in only a few vertebrates, including tentacled snakes, crocodiles, alligators, ducks, geese, star-nosed moles, monotremes, and big-brown bats [29].

L 235 – This section on function deserves its own section in the Discussion.

**This now has its own section in the completely rewritten Discussion.

L 250 – This is repetitive...See larger comment about the need for re-organization of the Discussion above.

** We have reorganized the discussion as suggested.

Figure 1 – Please give fish size. Distribution of “putative push-rod receptor complexes” should be used. Layers in the enlarged diagram (inset) are confusing. This diagram and labels (!) should include papillae next to the “complex”, basement membrane (separating epidermis and dermis), and the two greys are too similar to be seen easily.

** A scale bar has been added to the image of the full remora disc. Wording has been changed as suggested. We have added two additional images to illustrate the relative abundance of the putative push-rod mechanoreceptor complexes in the anterior and posterior lip and added labels to the first panel of the figure. Papillae not included in the inset because it is specifically a schematic diagram of the

mechanoreceptor complex to demonstrate its 3-dimensional structure.

Figure 2 – A lower magnification image should be added for context. Why differentiate between anterior and posterior lip regions? (This was not mentioned in the text, I believe). These are not very good SEM's. The skin does not appear to be clean (are there microridges on this skin?) and the surface structure of the surrounding skin cells appears to be distorted or obscured by “goop”.

** We have now added a stepwise increase in magnification to give a better view of the SEMs in perspective. We have removed the differentiation between anterior and posterior because the difference in abundance was better captured in the improvements to Figure 1. There are indeed microridges on the skin! This will be discussed in a different paper about soft tissue mechanics - the focus of this paper is on the finding of tactile receptive complexes in a fish.

Figure 3 – Nice figure, but first sentence should be “visualized by μ CT with phosphotungstic acid (PTA) staining.” Can't see the “dotted line”. “Mechanoreceptors” cannot be seen, only the putative “complexes”. What is a transaxial slice and how does it differ from a cross-section through the lip?

**Thank you! Text edited as suggested with regards to changes for “visualized by”, “mechanoreceptor complexes”, and “cross-section”. The dotted line has been enhanced for clarity.

Figure 4 – Title of figure needs to be more inclusive or the diagrammatic representation should be split into another figure. In addition, one section in each plane is not sufficient to illustrate the structure, which is presumably represented in the diagram.

** We have changed the title to be “Morphology and histology of the remora push-rod mechanoreceptor complex”. In addition we have substantially revised this figure to include comparative figures of the push-rod mechanoreceptor complex in monotremes, as well as new histological images to show the remora structure in more detail. Given the size of the structure it was not possible to capture in a single section (despite a great deal of painstaking trying).

Appendix B

Dear Editors,

We are delighted to hear that our manuscript submission entitled "Knowing when to stick: touch receptors found in the remora adhesive disc" was accepted with minor revision. Following is our revised manuscript and response on the requested corrections. We are extremely appreciative of the thorough and thoughtful insights provided by the two reviewers and Associate Editor Socha and are very grateful for their efforts and contributions. We have followed all their suggested recommendations (please see our replies in ****blue**** below) and hope you find our submission much improved.

Associate Editor's comments (Dr Jake Socha):

The authors have satisfyingly addressed the comments and concerns of both reviewers. Although there is still some question about statistical comparisons, the limited access to specimens and the tentative framing of the result (as 'putative') is justification for moving forward to publication. This paper provides the first evidence of push-rod-like mechanoreceptor complexes in fishes, and thus will serve as motivation for future studies to delve into the problem. Congratulations on a nice piece of morphological work.

One small edit to add: in line 45, change the text to "push-rod-like" (as in line 48).

****Thank you, we have made the correction in line 45 as requested.**

Reviewers' Comments to Author:

Reviewer: 1

Comments to the Author(s)

This revisions is much improved and the authors added great information to the morphology aspects (gorgeous figures!), but the quantification of these receptors is still lacking. There is one t-test on $n=3$; without the quantification this story seems superficial. Understanding the arrangement and densities might help infer function or mechanisms.

****Thank you, we are glad you enjoyed the revised morphological presentation! The reviewer should note that Figure 1 is an accurate representation of the relative arrangement and density of these mechanoreceptors. We would like to point out that in order to make a statistical comparison of density that would provide meaningful information regarding function, it would have to involve numerous individuals of all 8 species, as they have different host associations and therefore different shear environments. While this is certainly interesting, it is well outside the scope of this paper which is intended to detail the finding of these mechanoreceptors for the first time in something that is not a monotreme. And, with all due respect, knowing arrangement and density does not in itself offer any information regarding function or mechanisms, as so little is known about the hydrodynamic environment of remoras on different hosts in the first place.**